# Foraging Behavior and Pollination Efficiency of *Apis mellifera* L. on the Oil Tree Peony ‘Feng Dan’ (*Paeonia ostii* T. Hong et J.X. Zhang)

**DOI:** 10.3390/insects10040116

**Published:** 2019-04-25

**Authors:** Chunling He, Kaiyue Zhang, Xiaogai Hou, Dongbo Han, Shuaibing Wang

**Affiliations:** 1Forestry College, Henan University of Science & Technology, Luoyang 471000, China; zhangkaiyue95@126.com (K.Z.); handongbo1994@163.com (D.H.); wsb19960820@163.com (S.W.); 2College of Agriculture, Henan University of Science & Technology, Luoyang 471000, China

**Keywords:** *A. mellifera*, oil tree peony, visitation characteristics, visitation frequency, seed set

## Abstract

To solve the issue of insufficient pollinating of insects for the oil tree peony ‘Feng Dan’ (*Paeonia ostii* T. Hong et J.X. Zhang) and improve its seed set and yield, we conducted observations from 2017 to 2018 to investigate the relationship between honey bee (*Apis mellifera* L.) foraging behavior and diurnal activity. We compared the single-fruit seed set ratio among three flower types on the same plants of the oil tree peony, which flowered simultaneously, in three pollination areas (bee pollination, natural field pollination, and controlled pollination by pollinators) and in a net room under self-pollination, wind pollination and bee pollination. *Apis mellifera* exhibited short single visitations, long visitations to a single flower and repeated visits to flowers of the oil tree peony. The number of flower visits of *A. mellifera* was significantly and positively yet weakly correlated with the number of stigma visits (2017: r = 0.045, *p* < 0.05; 2018: r = 0.195, *p* < 0.01). The seed set of oil tree peony follicles in the *A. mellifera* pollination area was significantly higher than that in the natural pollination field area and the control net rooms. On the same oil tree peony plant with synchronous flowering, the percent seed set of follicles pollinated by *A. mellifera* at a high density was significantly higher than that resulting from wind pollination and self-pollination.

## 1. Introduction

Peony (*Paeonia suffruticosa* A.) is a perennial deciduous shrub belonging to the genus *Paeonia* L. and the family *Ranunculaceae* [1]. Peony, native to China, has been cultivated for over 2000 years and was introduced to Japan, Europe and the Americas during the Tang dynasty [2]. It has desirable ornamental characteristics and medicinal value and is an ideal oil plant [3,4,5].

The oil tree peony is a new type of woody oil crop and an important cash crop unique to China. It has been widely planted in Shandong, Henan, Gansu, Anhui, Hubei, Chongqing, Qinghai, Tibet, and other areas in China in recent years [6]. The oil tree peony is characterized by a high seed yield, high oil content (20–35%) and high fatty acid content (unsaturated fatty acids (UFAs) > 90%) [7]. In particular, the α-linolenic acid content in peony seed oil generally exceeds 40% [7,8]. Peony seed oil, in addition to being used as an edible oil, is widely utilized in cosmetics, medicine and other industries [9].

Peony is a typical insect-pollinated plant, and its main pollinators are bees (such as *Andrena carbonaria*, *Andrena stellaria*, *Lasioglossum simplicior* and *Bombus ignitus*) and beetles (such as *Hybovalgus bioculatus* and *Cetonia magnifica*) [10]. Both *Paeonia delavayi* F. and *Paeonia decomposita* F. rely on pollinators to increase their seed set [11,12]. The fruit set of self- and wind-pollinated ‘Feng Dan’ *(Paeonia ostii* T. Hong et J.X. Zhang) is extremely low [13]. 

Bees provide key ecological services for natural and agricultural ecosystems [14]. The total economic value of pollination services provided by bees and other insects worldwide amounted to €153 billion in 2008 [15]. The honey bee *Apis mellifera* L., one of the most widespread and common pollinators, can pollinate a variety of crops and boost yield [16,17]. At present, approximately 73% of the world’s cultivated crops, such as cashews, squash, mangoes, cocoa, cranberries and blueberries, are pollinated by some variety of bee; 19% by flies; 6.5% by bats; 5% by wasps; 5% by beetles; 4% by birds; and 4% by butterflies and moths [14]. Pollination by bees not only improves the seed set and quality of crops such as fruit, vegetable, nut and forage crops but also greatly facilitates the fruit yield and oil yield in oil crops including beans, sunflower, rape, sesame and cotton [18,19,20]. The positive effect of bees on pollination is attributed to their high pollination efficiency. Bees’ collection time, visitation times and pollen deposition are key factors for measuring their pollination efficiency in crops [21]. The measurement of pollinator visitation frequency can clarify the relationships between pollinators, pollinated crops and the seed set [22]. According to the literature, the visitation frequency of pollinators significantly affects the seed set of rape crops [23,24]. Oil tree peony is a superior woody oil crop; however, at present, little is known about the pollination efficiency of bees in the oil tree peony.

Therefore, we examined the foraging behaviors of *A. mellifera* on the oil tree peony ‘Feng Dan’ for two consecutive years, 2017 and 2018, mainly in terms of the time of the visit, visitation frequency, number of visits to a single flower, and diurnal activity, and their impact on the seed set of the oil tree peony. The results of this study may provide fundamental information for further exploiting the positive effects of bees on crop pollination and improving the yield of the oil tree peony.

## 2. Materials and Methods

### 2.1. Sampling Location

The experiment was conducted in an oil tree peony plantation (34°38′30″ N; 112°39′43″ E; 125.5 m) from April 2017 to August 2018 in the East Garden of Yibin District, Luoyang city. The East Garden, located in southeastern Luoyang city and owned by Luoyang Zhenguan Peony Planting Co., Ltd., is an important part of the peony garden industrial cluster zone with an area of thousands of square hectometers in Yibin District, Luoyang. The East Garden is one of the gardens in Luoyang where the oil tree peony is grown over a large area. The oil tree peony planted in the garden is ‘Feng Dan’, the planting area of which exceeds 106.7 hm^2^. ‘Feng Dan’ has entered the full fruit period, with a stock age of more than eight years. The primarily native bees include *A. mellifera*, *Xylocopa appendiculata*, *Andrena* spp., *Lasioglossum* spp., *Anthophora* sp. and *Eucera* sp.

### 2.2. Research Materials

In March 2017, three plots where the plants normally grew were randomly selected from the oil tree peony plantation of the East Garden in Luoyang. Two nylon net pollination houses were built in parallel in the east-west direction. Each net house was 45 m long from east to west and 8 m wide from north to south, with a 2.1-m-tall wall and a 3.2-m-high roof ridge. Each house was divided into three rooms (specifications: 15 m × 8 m), and nylon polyethylene with a mesh size of 1 mm was used as a screen.

As shown in Figure 1, the A, B, and C rooms were the areas with *A. mellifera* pollination, with one hive of honey bees (approx. 3000 worker bees, i.e., 25/m^2^) placed in each room. CK1 included the control rooms without bees, and CK2 was the field area with natural pollination. The same oil tree peony management routine was adopted in each experimental area. The *A. mellifera* bees were supplied by Luoyang Lilou Apiary.

The nets were installed before the plants flowered and removed after the bee pollination period to reduce the possible influence of the shade of the nets on the reproduction of the plants as much as possible. After the first peony flower opened in the net room, a hive of *A. mellifera* was placed in rooms A, B, and C. A beehive, 120 mm in height, was placed in the middle of each room. A sugar feeder (50% sugar content) was placed in each beehive, and a drinker (filled with drinking water) was placed next to the beehive. The sugar water and drinking water were replenished and replaced regularly.

### 2.3. Observation Indices

Visual inspection, photography and videography were used to observe the foraging characteristics of *A. mellifera* on the oil tree peony. The single visitation duration, visitation interval, total visitation duration per flower, total number of visits to a single flower per foraging flight, frequency of visits to a single flower per min, and number of visits to a stigma per flower visit by *A. mellifera* on the oil tree peony were observed using the stopwatch matching method [25].

The measurement indicators were defined as follows: 

Visitation frequency per min refers to the number of flowers that each honey bee visits per min. Single visitation duration is the time a honey bee takes to visit a flower, specifically, from the time the bee alights on the flower to the time it leaves. Visitation interval refers to the interval from the time a bee leaves a flower to the time the bee visits a flower again, including the time during which the bee temporarily leaves the flower it visits and stops mid-air to groom pollen and the time during which it flies from one flower to another. The total visitation duration per flower is the total time from the time a bee alights on a flower to the time it leaves the flower, which is the sum of the time it spends visiting a flower and the time during which it temporarily leaves the flower and grooms pollen mid-air. The total number of visits to a single flower per foraging flight refers to the total number of a bee’s visits followed by pollen grooming to the same flower during one foraging flight. The stigma visitation number is the number of visits to a stigma during a bee’s visit.

### 2.4. Observation of Foraging Behaviors

In 2017, we made observations for three consecutive days (15–17 April). Two 2 m × 2 m sample plots were selected in each bee net room, with six plots in total. Thirty flowers in full bloom were labeled in each sample plot. Every day from 8:00 a.m. to 18:00 p.m., we recorded the number of *A. mellifera* visiting the flowers in each plot within 10 min every 30 min. Similarly, we tracked and observed *A. mellifera* from 8:00 a.m. to 18:00 p.m. each day in each plot and recorded the visitation times and number of flowers visited by a bee in the sample plot. After a bee flew out of the sample plot, we started to track and record another bee entering the sample plot. Furthermore, the temperature, humidity and wind were recorded during different time intervals (every hour) using a hand-held meteorological instrument (Kestrel 3000; Skykomish, WA, USA).

We made observations for three consecutive days (from 16–18 April) in 2018. Two sample plots of 2 m × 2 m were set up in each bee net room, and 30 flowers were labeled in each plot. The foraging behaviors of *A. mellifera* on the flowers in each plot within 10 min every 30 min were recorded. The observation time spans were 9:30–11:45 a.m. and 13:30–14:30 p.m. (peak periods for visitation by *A. mellifera*, as discovered in 2017).

### 2.5. Observation of Pollination Efficiency

In the budding stage (when the buds began to soften) [26] of ‘Feng Dan’, 50 oil tree peony plants that bloomed simultaneously in rooms A, B and C were selected, and three buds at the same flowering stage were selected on each plant. The three buds from the same plant were pollinated with different methods: One bud was bagged with a sulfate paper bag for self-pollination; another bud was bagged with a nylon bag with a mesh size of 1 mm for wind and self-pollination (wind pollination); and the other bud was not bagged during blooming to allow pollination by bees within the net house (bee pollination). A total of 450 flower buds on 150 plants were labeled in three net rooms with bees as pollinators. All paper bags and screen bags were removed 15 d after the end of the flowering period. On 10 May 2017, three groups of green fruits pollinated with the three different methods on the same plant were picked from the three bee net rooms for statistical analysis of the seed set of the oil tree peony, with nine fruits in total. Fifteen green fruits were randomly picked from the three bee net rooms (five per room), the control rooms (CK1; five per room), and the field control area (CK2) for statistical analysis of the seed set of the oil tree peony in different pollination areas. Due to a cold spell in late spring in 2018, no data were collected for the same plants treated with different pollination methods. The seed set refers to the percentage of the number of ovules that developed into seeds in each follicle (the number of ovules ranged from 10–22, with an average of 17.24 ± 2.44).

### 2.6. Statistical Analysis 

Data are presented as the mean ± standard error and were analyzed with SPSS 24.0. The Pearson correlation method was used to analyze the correlation between the numbers of single flower visits and stigma visits by *A. mellifera* on the oil tree peony. Independent-samples *t*-tests were performed to compare the flower visitation behaviors of *A. mellifera* between different years. The homogeneity of variances was tested using Levene’s method, and the one-way analysis of variance (ANOVA) followed by a least significant difference (LSD) test was used for comparisons among the different treatment groups. In the case of heterogeneity, nonparametric Kruskal-Wallis one-factor ANOVA followed by a Bonferroni test was used. *p* < 0.05 and *p* < 0.01 were considered statistically significant and very significant, respectively.

## 3. Results

### 3.1. Foraging Characteristics of A. mellifera

When *A. mellifera* collects pollen, some bees fall directly onto the outer ring of the androecium and scrabble inward to collect pollen, while others scrabble on the surface of the androecium to collect pollen. Some bees directly alight on the inner ring and poke their head into the androecium (Figure 2A–D). On the same flower, bees may alight repeatedly to collect pollen several times. After collecting the pollen, bees sometimes crawl over the gynoecium or fly up and hover directly above the gynoecium to groom pollen (Figure 2E,F).

The oil tree peony is visited by *A. mellifera* from the beginning to the end of its flowering period. The visitation of the oil tree peony by *A. mellifera* is characterized by short single visits, long visits to a single flower, repeated visits to a single flower, and visitation of the same flower by several bees simultaneously (Figure 2A). A bee may alternate visits between adjacent flowers.

### 3.2. Foraging Behavior of A. mellifera on the Oil Tree Peony

There were no significant differences in the total visitation duration per flower (independent-samples *t*-test: *df*_(1,466)_ = 1.102, t = −1.086, *p* = 0.294), visitation interval (*df*_(1,1320)_ = 1.834, t = 0.878, *p* = 0.176), per-minute visitation frequency (*df*_(1,86)_ = 0.613, t = −0.358, *p* = 0.436), number of visits to a single flower per foraging flight (*df*_(1,394)_ = 2.538, t = 0.079, *p* = 0.112), number of bees visiting flowers during the peak visitation period (*df*_(1,50)_ = 1.554, t = 1.422, *p* = 0.218), number of stigma visits of *A. mellifera* (*df*_(1,192)_ = 3.206, t = 1.894, *p* = 0.075), and stigma visitation ratio (*df*_(1,192)_ = 0.538, t = 1.536, *p* = 0.468) between 2017 and 2018. Although a significant difference in the single visitation duration was observed (independent-samples *t*-test: *df*_(1,1042)_ = 20.282, t = −4.090, *p* < 0.001), this difference might be due to more rainfalls during the flowering season in 2018. The results are summarized in Table 1. In general, the data obtained in the two years were not significantly different; therefore, the data were pooled.

The Pearson correlation analysis demonstrated a significant linear relationship between the number of flower visits by *A. mellifera* on the oil tree peony and the number of stigma visits because the correlation coefficient was significantly different from zero (for 2017: R^2^ = 0.045, *p* < 0.05; for 2018: R^2^ = 0.195, *p* < 0.01).

### 3.3. Activity Patterns of A. mellifera on the Oil Tree Peony

Based on observations of the activity patterns of *A. mellifera* on the oil tree peony, when the flowers of the oil tree peony were in a half-open state between 7:00 and 8:00 in the morning, when the temperature was between 14.8–15 °C, a small number of *A. mellifera* began to fly out of the nest, while most of the bee visitation occurred after 8:30 a.m., when the temperature rose to 21–22 °C. The majority of bees stopped their visitation between 17:45 and 18:00 p.m., and a few stopped their visitation after 18:00 p.m. The peak periods of the oil tree peony visitation by *A. mellifera* were between 9:30 and 11:45 a.m. and between 13:30 and 14:30 p.m. The temperature during the peak periods was between 28.6 and 30.3 °C and between 29.1 and 31.6 °C, respectively (Figure 3). *Apis mellifera* exhibited more visitation activity when the weather was fair. In the morning (before 7:00 a.m.) or in rainy and windy weather, *A. mellifera* rarely went out to collect pollen.

Analysis of the collecting behavior of *A. mellifera* on the oil tree peony revealed that the number of peony flowers visited by *A. mellifera* (one bee) per min was between 1 and 11 (n = 168) during different periods in a day, with a mean of 2.43 ± 0.16. One-way ANOVA followed by an LSD test indicated significant differences among different intervals of time (Figure 4A). The total number of visits to a single flower per foraging flight was between 2 and 28 (n = 168) during different periods in a day, with a mean of 8.06 ± 0.46. Significant differences were also observed among different intervals of time (Figure 4B).

### 3.4. Pollination Efficiency of A. mellifera

Analysis of the seed set of flowers on the same plant of the oil tree peony treated with different pollination methods indicated that the average seed set of the oil tree peony with *A. mellifera* as pollinators (open pollination) was 70.02% ± 2.68% (*n* = 45); that by wind pollination (bagged with nylon mesh bags) was 14.48% ± 3.79%; and that by self-pollination (bagged with sulfate paper bags) was 3.65% ± 1.68%. The one-way ANOVA followed by an LSD test revealed that the open-pollinated group showed a highly significant difference in the seed set rate compared with the wind-pollinated group (*p* < 0.001) and the self-pollinated group (*p* < 0.001), although a significant difference was also observed between the latter two groups (*p* = 0.006) (Figure 5). 

Analysis of the seed set of ‘Feng Dan’ in different pollination areas revealed that the seed set of follicles pollinated by *A. mellifera* was 71.16% ± 2.22% (n = 74), that in the control net house was 28.17% ± 2.85% (n = 76), and that in the field area with natural pollination was 52.16% ± 2.60% (n = 70). One-way ANOVA followed by an LSD test revealed that the bee group showed a highly significant difference in the seed set rate compared with the blank control group (*p* < 0.001) and the field control group (*p* < 0.001), although a significant difference was also observed between the latter two groups (*p* < 0.001) (Figure 6).

## 4. Discussion

At present, for pollinator-dependent flowering crops grown in large areas for commercial purposes, *A. mellifera* bees are important pollinators used to alleviate pollination deficits [27,28,29]. This experiment demonstrated that the seed set of the oil tree peony ‘Feng Dan’ can be effectively enhanced by increasing the number of *A. mellifera* bees.

*Apis mellifera* bees display different visitation characteristics and pollination efficiencies on different plants [30,31,32,33]. Bees remember their foraging areas and plants. Generally, they do not return to the flowers on which they have already foraged [34]. However, they are accustomed to foraging and pollinating plants of the same species and show stability in plant pollination [34]. This study found that *A. mellifera* bees can visit up to 10 or 11 flowers per foraging flight. The visitation to one flower by these bees was rather long, and they repeatedly visited the same flower, with a maximum of 18 visits. 

The pollination efficiency of bees is closely related to their foraging behaviors and visitation duration on the pollinated plants and the number of visits; furthermore, it is also related to the amount of pollen carried by the bees and that settles on the stigma of the plant [30,33,35,36]. In this study, we determined indices related to the foraging behaviors of *A. mellifera* on the oil tree peony. However, the amount of pollen carried by *A. mellifera* after one foraging flight and after visits to a flower and the amount of pollen that settled on the stigma remain to be determined in the future. Furthermore, this study found that *A. mellifera* always groomed collected pollen above the stigma of the oil tree peony flower after pollen collection. The amount of pollen dropped on the stigma during collected pollen grooming also needs to be determined.

Departure from the hive by bees for collection is closely related to multiple factors such as temperature, humidity and irradiance, and the impact of temperature is far greater than that of irradiance, wind velocity and other factors [37,38,39]. The visitation of *A. mellifera* bees to strawberry flowers in the greenhouse begins at 9:25–9:40 a.m. (temperature > 15 °C), with a number of visited flowers per min of 2.38 ± 0.15 and a visitation interval of 6.0 ± 0.48 s [31]. Their visitations to cucumber and watermelon flowers begin after 7:20 in the morning (temperature > 22 °C), with a number of visited flowers per min of 8.0 ± 0.2 and 8.3 ± 0.2, respectively [40]. In this study, an investigation of the diurnal behaviors of *A. mellifera* on ’Feng Dan’ was performed during the day on sunny days. The number of visits to a single flower and the frequency of visitation during different time intervals showed significant differences, and both showed decreased activity at noon, when the temperature was the highest. The temperature at which *A. mellifera* showed the most active visitation on the oil tree peony in this study was consistent with those reported in the literature [30,40]. The peak period of the oil tree peony visitation by *A. mellifera* was between 9:30 and 11:45 a.m. However, whether the pollen viability and stigma receptivity of the peony are highest during this period warrants further study.

Pollination, the process in which pollen is transferred from the anthers of the stamen to the stigma of the pistil, is critical for flowering plants. The seed set is usually used as an indicator of the success of pollination. The fruit and seed set refer to the ratio of the plant’s flowers or ovules ultimately developing into fruits or seeds, respectively. Because of the natural shedding of fruit, inadequate pollination, herbivore interference or management problems, it is difficult to achieve 100% fruit or seed set [41]. For crops that rely on pollinators for pollination, pollination efficiency is directly affected by the abundance of pollinators [42,43]. ‘Feng Dan’, characterized by a high seed yield and seed oil content and good oil quality, is the main cultivar used in China’s oil tree peony industry; however, it is a cross-pollinated plant with a low (potentially zero) self-pollination seed set [13,44]. From 2017 to 2018, the seed set of ’Feng Dan’ pollinated by *A. mellifera* was significantly higher than that in the control net house and field area with natural pollination. In this study, we set up a net pollination house to increase the pollination of ’Feng Dan’ by western honeybees. The results of our study show that the seed set of ’Feng Dan’ is significantly improved by bee pollination and that *A. mellifera* is an effective pollinator of the oil tree peony ‘Feng Dan’; however, the impact of pollination on the yield and quality of peony seeds requires further study.

## 5. Conclusions

The seed set of oil tree peony follicles can be significantly increased by *A. mellifera*. The results of our research indicate that *A. mellifera* is an effective pollinator of the oil tree peony and thus can greatly boost its yield.

However, this study may suffer from the limitation of a small sample size. To further validate the findings of this study, studies based on a larger sample size remain to be conducted. In addition, the findings of this study were obtained with net houses. The pollination effect of *A. mellifera* in the field environment and the optimal density of bees in the field environment need to be explored.

## Figures and Tables

**Figure 1 insects-10-00116-f001:**
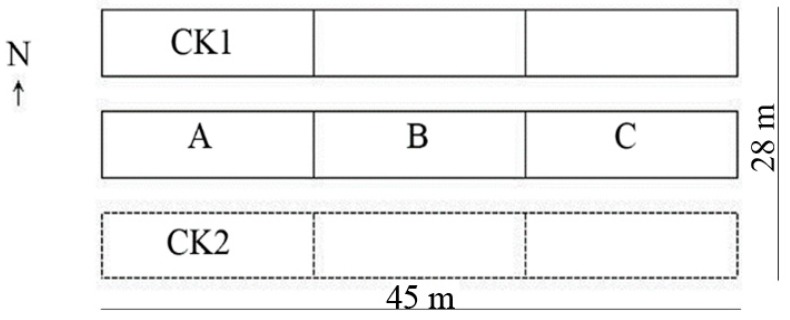
Plan graph of the net pollination houses built in the oil tree peony (‘Feng Dan’) plantation. A, B, and C are the *A. mellifera* net rooms. CK1 includes three net rooms without bees. CK2 is the field area with natural pollination.

**Figure 2 insects-10-00116-f002:**
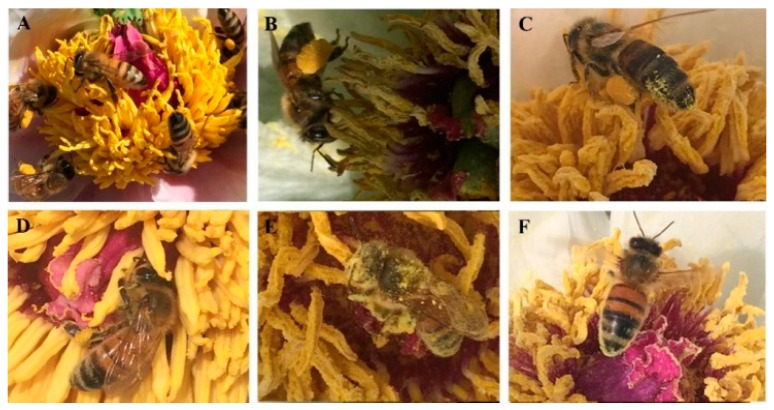
Foraging characteristics of *A. mellifera* on the oil tree peony ‘Feng Dan’. (**A**) Multiple bees collecting pollen on the same flower; (**B**) a bee collecting pollen from the outer ring of the androecium; (**C**) a bee crawling from the inner ring of the androecium toward the outer ring; (**D**) a bee collecting pollen from the inner ring of the androecium; (**E**) nearly the whole body of a bee contacts the stigma; (**F**) a bee grooming pollen above the stigma.

**Figure 3 insects-10-00116-f003:**
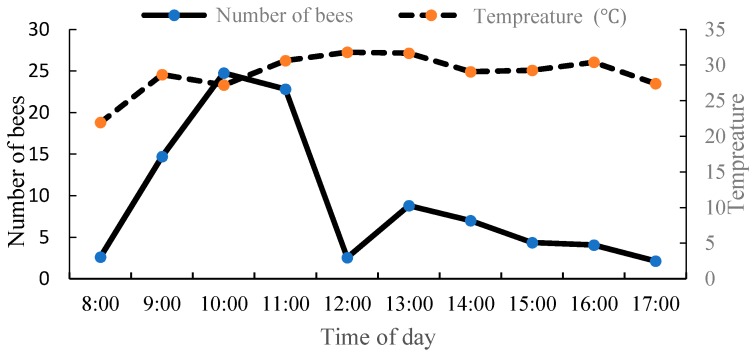
Diurnal activity of *A. mellifera* on the oil tree peony ‘Feng Dan’.

**Figure 4 insects-10-00116-f004:**
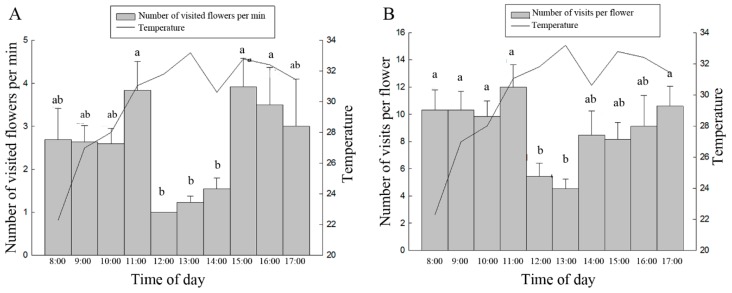
Collecting behavior of *A. mellifera* on the oil tree peony ‘Feng Dan’ in different time periods. (**A**) Number of flowers visited by *A. mellifera*; (**B**) number of *A. mellifera* visits to oil tree peony flowers. Notes: The data in the figure are presented as the mean ± standard error, and different letters above the columns in the figure indicate a significant difference at the 5% level (one-way ANOVA followed by an LSD test).

**Figure 5 insects-10-00116-f005:**
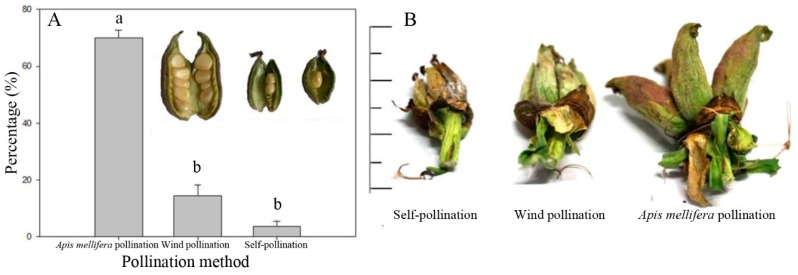
Seed set of the oil tree peony ‘Feng Dan’ treated with different pollination methods (**A**) and clustering follicles of the oil tree peony (**B**). Notes: Data in the figure are presented as the mean ± standard error. Different letters above the columns in the figure indicate a significant difference at the 5% level (one-way ANOVA).

**Figure 6 insects-10-00116-f006:**
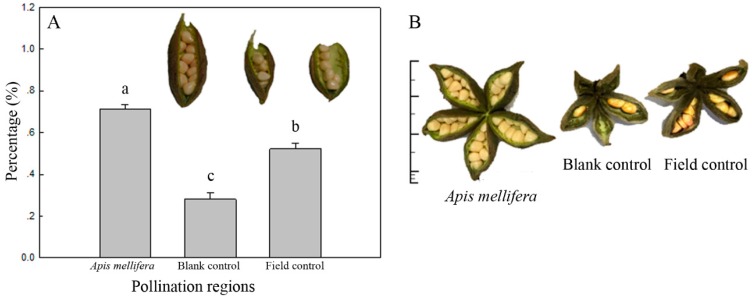
Seed set of the oil tree peony ‘Feng Dan’ in different pollination areas (**A**) and clustering follicles of the oil tree peony (**B**). Data in the figure are presented as the mean ± standard error. Different letters above the columns in the figure indicate a significant difference at the 5% level (one-way ANOVA).

**Table 1 insects-10-00116-t001:** Foraging behaviors of *A. mellifera* on the oil tree peony ‘Feng Dan’ in different years.

Parameter	2017	2018
Single visitation duration (s)	3.48 ± 0.11b	4.33 ± 0.18a
Total visitation duration per flower (s)	42.36 ± 2.62a	46.57 ± 2.84a
Visitation interval (s)	5.23 ± 0.13a	5.06 ± 0.14a
Number of flowers visited per minute	1.46 ± 0.55a	1.50 ± 0.59a
Number of visits to a single flower per foraging flight	5.15 ± 0.23a	5.12 ± 0.30a
Number of bees visiting flowers	21.23 ± 2.17a	17.08 ± 1.33a
Number of stigma visits	0.92 ± 0.14a	0.63 ± 0.11a
Stigma visitation ratio (%)	7.79a	8.13a

Notes: The data in the figure are presented as the mean ± standard error, and different letters in the same columns indicate a significant difference at *p* < 0.05 (independent-samples *t*-test). Stigma visitation ratio refers to the percentage of stigma visits out of the total number of visits to a single flower.

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
