# Peer review of "Foraging Behavior and Pollination Efficiency of Apis mellifera L. on the Oil Tree Peony ‘Feng Dan’ (Paeonia ostii T. Hong et J.X. Zhang)"

_insects, 2019, doi:10.3390/insects10040116_

Round 1
Reviewer 1 Report
Manuscript Review: Insects/458863
Foraging behaviour and pollination efficiency of Apis mellifer L. on oil tree peony
Overall this is a good piece of research, novel in it's content and contextual to this journal and the readership. I hope that my comments below will serve to help polish it and bring out the best and most important aspects of the research.
The paper could have greater impact, however, if the focus were firmly placed on the concept expressed in line 276-277, diminishing all the distracting year-on-year results and focussing instead on the actually matter in hand - that is, improving pollination efficiency and hence seed set. Too much of the length of the results section is dominated by the year-on-year analysis, which really only demonstrates consistency of the research and of the pollination syndrome. As such, it doesn’t really add anything, whereas if this is diminished in length, the actual value of the paper will shine better. Hence, I recommend removal of figure 3 entirely and vigorously shortening the text that goes along with it (lines 161-194) to simply state that over the two years there was no significant change and that the data was consequently pooled. Then the remaining analysis will have strength and clarity. Of special concern are the apparent misapplications of ANOVA inherent in some comparisons (see below). These need to be corrected to enhance the validity of what is being concluded.
Title: Good, clear and focussed, directing the read to the content. US spelling; behavior versus behaviour?
Abstract: Write in full the generic name at the beginning of sentence, that is Apis no A.
The correlation is weak according to the low R2 value. It would be preferable to state: "The visitation time of A. mellifera was significantly and positively, yet weakly, correlated with its stigma visitation time (R2=0.195, P<0.01)."
Introduction:
Throughout: replace net rooms with growth tunnels (this is the more commonly used phrase for such structures).
Line 41 (end of the sentence): Should pollination deficit read pollinator deficit? The emphasis is then placed on the lack of insects.
Line 59 (end of the line): To avoid starting the next sentence with a conjunctive, and to link the two concepts, change the full stop to a comma and begin however with a lower case.
Line 61: same reason: rephrase "For two consecutive years, 2017 and 2018, we therefore, examined the pollination efficiency of A. mellifera on oil tree peony."
Materials and Methods
Line 69: delete and between latitude and longitude and replace with a semicolon ;
Line 69: delete with an altitude of (it is self evident that this is an altitude).
Line 73: conver mu to SI units (i.e. 6.5M m2)
Line 83 & 87: replace box with hive
Line 87: make flower plural i.e. flowers
Caption to Figure 1 should explicitly explain the left hand diagram.
Line 99: change "that each bee can visit per min” to "that each bee visits per min”
Line 116: change retrack to track
Line 117: meaning is not clear because of conflict between “at the same time” and "at different time intervals”
Line 120: delete same
Lines 140-145: this section needs to more transparent. For example, was a homogeneity of variance established, if not, then what procedure was followed; furthermore, were samples pooled year-on-year? Why use Excel and Sigmaplot, when SPSS can conduct all the tests that were undertaken, is more robust and produces document ready plots. Be sure to use the correct analysis for the hypotheses that are set up.
Results
Throughout: t-test should be hyphenated
Section 3.2 Foraging behaviour of A. mellifera on oil tree peony: This section is a distraction; I fail to understand the purpose of comparing two years of activity, other than to establish consistency and enable pooling of the data for further analysis. It is a valid test, but it need not be so explicitly entertained and space consumptive. This section could be considerably reduced by stating that, other than single visit times, all comparisons for foraging behaviour as outlined in section 2.3, were not significantly different across years. If it is necessary to demonstrated the results, then tabulate them and delete figure 3, as it really doesn’t serve a purpose, and the focus can be redirected to the foraging activity and seed set efficiencies of A. mellifera behaviour on oil tree peony. What really matters, is not how the bees differ or not over subsequent years (unless you are researching the reasons for these differences), but how efficient the bee pollination is in comparison to other pollination syndromes.
In contrast section 3.3 provides clear and valid observations.
Lines 191 - 194: this is not strictly true and the plot Fig 3F is inappropriate, as the Pearson correlation analysis is an expression of a linear relationship. A more appropriate conclusion is: The Pearson correlation analysis demonstrated that there is sufficient evidence to conclude that there is a significant linear relationship between the number of visits by A. mellifera on oil tree peony and stigma visitation times, because the correlation coefficient is significantly different from zero (for 2017 R2=0.045, P< 0.05; for 2018 R2=0.195, P<0.01).
Line 199: this is a well known diurnal behaviour for bees, and yet these results are not linked in the discussion to the vast and relevant literature about foraging patterns in Apis mellifera.
Section 3.3 Activity patterns of A. mellifera on oil tree peony: it is not clear to me whether these activities are out-doors or inside the growth tunnel. This is relevant in context of the comment about conditions outside the tunnel (Lines 206-209 and Figure 5). How do the observed differences in conditions relate to bee activity?
Lines 214 - 221: the analysis needs greater clarity. Firstly what is being compared with number of flowers visited? Is it visits per hour? This is not clearly established. More importantly, an ANOVA analysis results in an F-value and significance that only indicates a difference between one or more pairs of crosswise tests exists. If the result is significant, it therefore requires to be followed by a post hoc test to indicate which cross-wise pairs are significant and which are not. No such post hoc test is indicated and it was also not established that the assumption of homogeneity was upheld prior to the analysis. The reference to the ANOVA conflicts with the caption of Figure 6, which refers to independent sample t-tests. This suggests confusion about the role of t-tests comparing two means and ANOVA comparing more than two means. T-tests cannot be used as post hoc tests to the ANOVA analysis - that would increase the risk of a Type I error as the t-test is not robust enough to measure the variance between multiple data sets greater than two.
Temperature plots between figures 5 and 6 are inconsistent (figure 5 demonstrates a steady rise towards noon, then a gradual fall; figure six demonstrates fluctuations at noon). If these data come from the same source, then the plots should be identical. The y-axis scales may need adjustment to make this clear.
Lines 227-232: can reliably be included in the general statements regarding annual comparisons in section 3.2. Figure 7 serves no purpose and could be deleted.
Lines 238 - 243: This section suffers the same misapplication of ANOVA as described for Lines 214 - 22. Furthermore, while not invalid as a comparison, it serves little purpose experimentally, unless to confirm what is already known - that is that peony is insect pollinated, not selfed and not wind pollinated. If the purpose is to demonstrate or clarify some confusion in this matter, fair enough, but I don’t see it and this test appears to be included without rationale simply because it can be.
Lines 248 - 252 and Figure 9: this is the real core of your research! There is, however, no need to separate the two years, as you already established that there were no significant differences between years.
Lines 257 - 265: the same comments apply to the misapplication of ANOVA as above. In fact, accepting that the year-on-year data can be pooled, this should be a pooled t-test, not an ANOVA.
Line 267: needs a citation.
Line 288: clarify at a time, because the way this is currently written suggests that a single bee can be in ten flowers at the same time - which is of course nonsense. Perhaps you mean 10 or 11 flowers per foraging flight?
Lines 290-292: Delete this last sentence, because not only does it diminish the impact of the current findings, but there also appears to be no such connection.
Line 312: replace nest with hive
Lines 322-323: repetition; delete "When collecting peach pollen, A. mellifera can determine the degree of pollen viability and are inclined to visit flowers with greater pollen viability [37]"
Line 333: end with comma, lower case however ….
Line 336: change control net room to control tunnel
Line 336-338: redundancy; delete, "On the same oil tree peony plant, the seed set of flowers pollinated by bees was significantly higher than that of wind- or self- pollinated flowers."
Line 340: delete “preliminary” - it undermines the strength of your conclusions.
Line 341: end with comma, lower case however ….
References and end comments: all appear to be in order.
Reviewer 2 Report
Overall assessment: The authors are aiming to answer an important and applied question: does increasing honey bee abundance affect the reproduction of a species of oil tree peony (Peony suffruticosa) in an agricultural setting? It is a straightforward question with a relevant and important answer. The authors used a very realistic field site to collect data and collected an impressive amount of data.That said, the paper could use some more attention to what background information is needed in the introduction, and the discussion should focus on their results in the context of the study question rather than introducing a massive amount of new and speculative information. The data are also analyzed in a way that makes it unclear if the tests were truly independent, and the authors could do a better job of justifying what metrics they used and why. A few high-level comments are presented below, followed by specific line-by-line comments. The explanations of the methods could use some clarification, and it was very difficult to sift through the paragraph defining the measurement indicators.When describing the net rooms, the long 45 m x 8 m units and the smaller 15 m x 8 m units should have different labels rather than both being called net rooms. Many of the measurement indicator terms were so similar and words like “time” used so frequently that it was difficult to maintain a knowledge of them while reading the paper; “single visitation time” versus “visit to a single flower”, for example, was challenging to maintain. The methods for selecting sample plots and groups of fruits were also unclear. With such a large amount of data being collected for a multifaceted study that assesses foraging behavior, three pollination methods, and three pollination areas, the need for clarity cannot be stressed enough. The authors seem to use fruit set and seed set interchangeably throughout the paper, stating multiple times that fruit set and seed set had been boosted by the presence of A. mellifera; however, I’m not sure that they tested for fruit set. In my experience, fruit set refers to the number of fruits produced per plant and seed set refers to the number of seeds produced per fruit. While they do mention in line 131 that they collected three groups of green fruits for “analysis of fruit set of oil tree peony treated with different pollination methods,” all the results provided in section 3.4 discuss “seed set.” The discussion section then asserts that the “fruit set” was significantly improved, and finally, in the conclusion, the authors revert to using the term “seed set” as being significantly increased. The authors should be more consistent and clear with the terminology they are using. It would be worth it to include in the research whether oil tree peony (Peony suffruticosa) is a plant that will divert resources from buds with fewer pollinated follicles to buds with many more pollinated follicles. If so, it’s possible that could have an effect on seed set, size, and quality during the comparison between bee pollination, wind pollination, and controlled pollination under the current design of selecting three buds on the same plant. Perhaps further studies could subject a whole plant to each treatment to avoid that potential. Further experiments designed to assess A. mellifera impacts on fruit set and seed set are merited. Given the highly realistic setting of this study, some suggestions for applications of the paper’s important conclusion of increased seed set due to bee abundance can be valuable to the oil tree peony industry in terms of improving seed oil yield. Would the authors suggest that farmers set up net rooms each stocked with a bee hive during the flowering season? This study employs a high ratio of bees to plants, approximately 3,000 worker bees for each 15 m x 8 m net room. Would the seed set percentage be as high if bee boxes were simply placed in an open field? Minor comments. Line 11: Are there different common names for the various species? I was only able to find “tree peony”, which seemed to refer to multiple species. Perhaps clarification would be helpful? Line 19: this hardly seems like the most important takeaway from the study, which is that increasing pollinator density dramatically increased seed set and yield and could be an effective way to increase farming profits, unless I am mistaken. Line 29: What is the wild bee community there like? Is the honey bee native? How many other species are native? Could the authors please characterize the bee fauna and community? What are the native pollinators of the oil tree peony? Line 35: please provide a citation for fat contents (or clarify that the citations are the same as in the next sentence) Line 38: What kinds of bees? Again, could the authors please go into more detail on the natural pollination strategies of this plant? Line 41: Is this is the results of this study or a previous one? If it is from this study, this belongs in the abstract or results rather than the introduction. If it is from a previous study, this statement needs a citation. Line 45: Honey bees? Or does this include non-honey bee bees? “Bees” could mean a lot of things that could have dramatic consequences for this paper, so I think it would be wise to clarify here and throughout (e.g., lines 52 and 60). Line 59: Please clarify. Oil tree peony is a better oil crop than herbaceous crops? Or this species of peony is better than other woody species? Line 61: I am not certain that pollination efficiency is what was measured. I feel like it was more about pollination services. I could be incorrect, but I feel that you would need to specifically look for efficiency the authors would need to look for pollination (in terms of fruit/seed set) as a function of visitation, not just number of bees visiting (since the visitation may or may not result in pollination, so the pollination efficiency would also not be measured). Line 63: ‘number’ of what? Please clarify Line 63: diurnal activity? Versus nocturnal? Or activity throughout the day? Please clarify Line 66: the basis of a shortage of edible vegetable oil in China has yet to be established. This needs a citation at least, if not a broader introduction earlier on as a way to set up the importance of the study. Line 81: Perhaps a different term other than “room” should be used for the smaller units to avoid confusion further on. Line 82: Is there any fear that the shade from the fabric affected the reproduction of the plants? Is there any way to address that as a potential confounding factor? Figure 1. Please be more descriptive in the figure caption. What are the codes? Line 95: The stopwatch method could use some further explanation. What was done exactly? And the citation is for Xylocopa, does it translate to honey bee work? Line 104: “. . . one flower to another” and comes back? Total visit duration might be a better term for this interval. This whole section is difficult to sift through with so many similar terms. Line 107: each time = each stopwatch duration? 10 minutes as described in Line 113? Line 110: does this mean that there were two plots total and each one was observed twice for three days? So 12 total observation periods? Or 12 per “room”? Please clarify the sampling scheme. Line 114: how often did you record them for 30 minutes? Line 117: Be more specific than at different time intervals. Every hour? Line 120: I think this could be streamlined to show what is the same and what is different between the years. As it is now I am uncertain what happened and what was different between years. Line 124:How were the buds selected? Line 125: does peony divert resources to flowers that receive the most pollen? In other words, does it make sense to pick flowers on a single plant versus either picking one flower randomly per plant or treating on the whole plant level? Line 130: please be more specific than ½ month. 14 days? Line 130: How many ovules are there per ovary? Line 131: How many fruits were picked in each of these groups? Line 134: Which bee net rooms? I, E, A? Line 142: I have major concerns over the use of t-tests following the ANOVAs, unless they are truly independent (in which case the methods need to be very clear). How were the t-tests made independent? Did you set up apriori linear contrasts? Did you use a Bonferroni corrected alpha? If the t-tests were not truly statistically independent, i would suggest using a post hoc Tukey HSD test to determine differences among the groups. Figure 2. The photos are lovely could use some interpretation. Please describe what the reader is seeing in each panel. Line 163: please indicate what metric of error is being shown. SD? SE? CIs? Line 165: please list the degrees of freedom, which would help assess if the t-tests were independent. Figure 3. Please change the x-axis to “Year” and only include the axis label on panels E and F. Line 170: I suggest changing the prose to “different letters above the columns indicates a significant difference with P < 0.05” Line 172: what is the difference between “single visit time” and “visitation time on a single flower”? Line 187: over the course of 10 minutes? Or over the course of the bees total visitation time? Line 193: Where is the stigma visitation time data? Figure 4: Given that the temperature never got below 20, it seems that starting the y-axis at 20 would help see the pattern a great deal. Likewise, might it be beneficial to show a scatterplot of activity and temp so we can see that relationship more easily? That said, it does not appear that there is a relationship with # of bees and temperature at all, so is this data worth presenting unless the point is being made that foraging behavior is not affected by temperature here? Figure 5: I do not find this figure helpful. I am not sure what patterns I am supposed to see here. Lines 214-220: are the floral resources equal over the course of the day? Does the daytime foraging activity respond more to temperature or other abiotic factors? Or to resource availability? Or something else? Please specify which year this data applies to. Figure 6: I am not sure what I am supposed to get from the temperature lines. Also, what is ‘mean’ time of day? Line 226: Again, I think the authors need to be very specific about how they determined differences among the groups. Using t-tests is a bit of a red flag. How did the authors maintain statistical independence? How are the authors preventing type I error? Figure 7. Please change “number of pollinations” to “number of visits”. Determining pollinations would require determining that there was fertilization. Line 236: same comment as before about the P < 0.05 Figure 8. Is there no pollination by non-honey bee bees? Figure 8 B. I love these photos. They are so descriptive. Line 251: I think you may need a more complicated model structure than a one way ANOVA. The degrees of freedom seems wrong given the study design. Perhaps a nested ANOVA or a mixed model would be a better approach. Figure 9. Please change x-axis label to “Pollination method” Line 275: fruit set only? Not seed set? Line 279: citation needed Lines 280 - 285: this feels like material that should be in the introduction, not the discussion. Line 287: I am not certain why foraging for more viable pollen is relevant. If I am missing something, maybe it could be clarified? Lines 288 - 292: This appears to be the most relevant portion of this discussion so far. The rest should be streamlined or eliminated. Lines 293 - 338: Some of this is very speculative and the information, in general, feels like material that should be in the introduction, not the discussion. Line 325: here and elsewhere the authors could consolidate the next steps. Currently, the discussion feels like speculation and introduction materials capped off with a next step. Line 344: the conclusion here is excellent, although could be lengthened slightly.
Round 2
Reviewer 1 Report
This is now an excellent paper - which I recommend for immediate publication.Thank you for your polite responses to my comments, which I am glad you found useful; I am grateful to have had the opportunity to assist you in bringing this valuable research to publication.
Author Response
Response: Thank you for your positive comment on our revision. Your constructive comments have been of great benefit for the enhancement of hte quality of the work. Thank you.
Reviewer 2 Report
Overall assessment: The authors have made major changes and progress toward clarifying their manuscript, and the result is a much more clear description of their study. The authors did a good job of streamlining the introduction and discussion sections by removing unrelated or repeated descriptions, and their clarifications of the measurement indicator terms and sampling methods were helpful in understanding the experimental design.These changes are much appreciated and the manuscript is greatly improved as a result. The one change that did not seem to address the previous concern was identifying in their research whether this species of plant will divert resources toward fruits that have been pollinated, thereby biasing their results to find a difference (line 134). My concern is that having different treatments on the same plant might have some confounding factors since the plant could divert resources from buds with fewer pollinated follicles to buds with more pollinated follicles, potentially exaggerating apparent differences in seed set among the pollination techniques. We greatly appreciate the clarification of the statistical methods. Minor comments. Line :
Figure 1. Could the authors use a GIS image or satellite map to make this easier to interpret? Maybe add dimensions?
Author Response
Overall assessment:
The authors have made major changes and progress toward clarifying their manuscript, and the result is a much more clear description of their study. The authors did a good job of streamlining the introduction and discussion sections by removing unrelated or repeated descriptions, and their clarifications of the measurement indicator terms and sampling methods were helpful in understanding the experimental design. These changes are much appreciated and the manuscript is greatly improved as a result.
The one change that did not seem to address the previous concern was identifying in their research whether this species of plant will divert resources toward fruits that have been pollinated, thereby biasing their results to find a difference (line 134). My concern is that having different treatments on the same plant might have some confounding factors since the plant could divert resources from buds with fewer pollinated follicles to buds with more pollinated follicles, potentially exaggerating apparent differences in seed set among the pollination techniques.
Response: Thank you for your concern and your comment is worthy reconsidering. The oil tree peony ‘Feng Dan’ is a perennial deciduous shrub. At 8 years, ‘Feng Dan’ has already entered the full bearing stage. At this stage, ‘Feng Dan’ has 6-9 end-bud flowering branches, normally, which grow synchronously and blossom at the same time period. In this study, we selected the buds that blossom at the same time period from the end-bud flowering branches in basically unanimous growth of the same plant to investigate the effect of different pollination methods on the seed set of the plant. Indeed, as you remarked, such a selection might bias the results obtained in this study (although slightly, we think). However, even the selection from different plants cannot completely exclude all confounding factors. Compared with the latter selection, we believe the former selection would be better, as the nutrients for the buds were provided by the same plant, at least. In addition, the purpose of this study was to investigate the effect of different pollination methods on the seed set of ‘Feng Dan’. If it is the case, that is, the plant could divert resources from buds with fewer pollinated follicles to buds with more pollinated follicles, it would be another angle to attest that bee pollination had higher pollination efficiency than any of other pollination methods. Therefore, such a factor would not influence the conclusions drawn from the experiment negatively.
We greatly appreciate the clarification of the statistical methods.
Response: Thank you for your positive comment.
Minor comments.
Line :
Figure 1. Could the authors use a GIS image or satellite map to make this easier to interpret? Maybe add dimensions?
Response: Thank you for your comment. We apologize that we did not have a GIS image or satellite map to show the plan graph. However, following your comment, we have integrated dimensions to the original Figure 1 as follows: